# Evaluation of Dynamic Properties of Sodium-Alginate-Reinforced Soil Using A Resonant-Column Test

**DOI:** 10.3390/ma14112743

**Published:** 2021-05-22

**Authors:** Seongnoh Ahn, Jae-Eun Ryou, Kwangkuk Ahn, Changho Lee, Jun-Dae Lee, Jongwon Jung

**Affiliations:** 1School of Civil Engineering, Chungbuk National University, Cheongju 28644, Korea; copoi2212@chungbuk.ac.kr (S.A.); jaeeunryou@chungbuk.ac.kr (J.-E.R.); akk@chungbuk.ac.kr (K.A.); 2Department of Civil Engineering, Chonnam National University, Gwangju 61186, Korea; changho@jnu.ac.kr; 3Department of Civil Engineering, Semyung University, Jecheon 27136, Korea; jdlee@semyung.ac.kr

**Keywords:** ground reinforcement, resonant column, biopolymer

## Abstract

Ground reinforcement is a method used to reduce the damage caused by earthquakes. Usually, cement-based reinforcement methods are used because they are inexpensive and show excellent performance. Recently, however, reinforcement methods using eco-friendly materials have been proposed due to environmental issues. In this study, the cement reinforcement method and the biopolymer reinforcement method using sodium alginate were compared. The dynamic properties of the reinforced ground, including shear modulus and damping ratio, were measured through a resonant-column test. Also, the viscosity of sodium alginate solution, which is a non-Newtonian fluid, was also explored and found to increase with concentration. The maximum shear modulus and minimum damping ratio increased, and the linear range of the shear modulus curve decreased, when cement and sodium alginate solution were mixed. Addition of biopolymer showed similar reinforcing effect in a lesser amount of additive compared to the cement-reinforced ground, but the effect decreased above a certain viscosity because the biopolymer solution was not homogeneously distributed. This was examined through a shear-failure-mode test.

## 1. Introduction

The damage caused by earthquakes is increasing worldwide. To mitigate it, seismic design of structures is mandatory [1]. Geotechnical engineering seeks to improve ground strength through physical or chemical methods, not only for seismic design but also for general stability [2]. Cement is the key ingredient in the most commonly used method for reinforcing the ground. Mixing cement with the ground can ensure strength at a low price [3]. However, when cheap cement and lime materials are used, the slaking and coagulation reaction of calcium hydroxide occurs, leading to the formation of ettringite. This increases the pH of the soil, impeding vegetation growth and accelerating soil degradation [4,5,6]. Moreover, it has been reported that CO_2_ emissions related to the production and use of cement amount to 3.4% of global consumption and 8~10% of total consumption [7,8]. Therefore, it is necessary to seek new eco-friendly and renewable materials that can increase the strength of the ground while reducing the use of cement. The MICP (microorganism-inducing biological polymer) and biopolymer reinforcement methods especially show potential in this regard [9]. In BPT (biopolymer treatment), dry powder composed of a biopolymer (e.g., xanthan gum, gellan gum, β-glucan, or chitosan) is mixed with water and then with soil and allowed to solidify. BPT increases erosion resistance, strength, and water-tightness [10,11,12,13,14]. Rigidity is improved by direct ionic bonding between biopolymers and microparticles or by the formation of a continuous biopolymer matrix [15,16,17].

The shear modulus and damping ratio, which are dynamic properties, are important parameters in seismic design of the ground under dynamic load. Below the linear limit strain, the shear modulus and damping ratio are nearly constant regardless of the strain; in this strain range, the shear modulus is at its maximum and the damping ratio is at its minimum [18]. Above the linear limit strain, the shear modulus and damping ratio show nonlinear behavior, and begin to decrease and increase, respectively. Parameters that affect the dynamic properties of the ground include confining pressure, saturation, pore ratio and soil structure [19]. Seed and Idriss determined normalized shear modulus and damping ratio curves according to the strain range for pure sand and clay [20].

Kim et al., Lee et al., Chang et al., Im et al. and others have compared and analyzed the properties of various sands, including Jumunjin Standard sand in Korea [2,21,22,23]. The dynamic properties of ground reinforced with waste tire rubber, bentonite, cement, and biopolymer have also been studied [2,22,24,25]. The results depended on the type of reinforcement material. When cement and polymer were mixed, the maximum shear modulus of the ground increased, and the linear range of the normalized shear-modulus curve decreased (Figure 1a) [2,25]. When the fraction of bentonite [24] or waste tire rubber [22] was increased in the sandy soil, the linear range of the normalized shear modulus increased (Figure 1b). However, the minimum damping ratio increased regardless of the type of reinforcement material (Figure 1c) [2,25,26]. In addition, as the confining pressure increased, so did the maximum shear modulus, whereas the damping ratio decreased, regardless of the composition of the mixed material.

Two of the materials in these studies, xanthan gum and gellan gum [2], are in the spotlight as renewable materials that reduce the amount of cement used. Chang compared the compressive strength and elastic modulus of soil treated with xanthan gum and gellan gum to that of soil treated with cement [17,23]. Chang and Cho also compared the economic efficiencies of the β-1.3/1.6-glucan polymer and cement [15]. However, sodium alginate mixed with soil has not yet been studied that is a substance extracted from marine brown algae. Alginate is used in the food, medical, and textile industries, and has gelling, viscous, and stabilizing properties with its ability to retain water [27]. When soil and sodium alginate are mixed, the permeability decreases and the resilient modulus increases [28]. Since the properties of this polymer have not been measured, exactly how much the effect of reinforcing the dynamic properties of the ground according to the polymer content appear cannot be known. In this study, the viscosity of sodium alginate solution was measured at various concentrations, and the change of the dynamic properties of the reinforced ground using sodium alginate with various viscosities and confining pressures was compared with that of ground reinforced using cement. Also, the viscosity-dependent characteristics of sodium alginate were confirmed through a shear-failure-characteristic test.

## 2. Materials and Methods

### 2.1. Materials

Weathered residual soil, which is the most widely distributed in South Korea and widely used as a ground composition and construction material, was taken at the field (Cheongju, Korea) and used. Table 1 shows the properties of the soil sample. The soil was classified as poorly graded sand (SP) in the unified soil classification.

Sodium alginate (SA) biopolymer (MP Biomedicals, LLC; Solon, OH, USA) and cement were used as mixing materials in studying the dynamic properties of the reinforced ground. The cement was KS L 5201 Portland cement. The biopolymer was a material made in powder form using materials that can be extracted from nature: the SA was made through alginic acid from the cells of algae, the brown seaweed. Alginic acid is composed of mannuronic acid and L-guluronic acid (Figure 2); depending on the ratio, elastic or hard brittle gels can appear. Because of these viscoelastic properties, SA is of industrial interest: applications being researched include heavy metal adsorption and edible or pharmaceutical products. Possibly it can be applied as a substitute for low-cost industrial materials such as Styrofoam and construction material or used to add value to other materials; it may also be useful in purification and polymer synthesis technology for reduction of environmental pollution [29]. In the present study, viscoelastic SA was dissolved in water and used as a reinforcing material to increase the strength and rigidity of the ground. The material properties of SA are shown in Table 2.

### 2.2. Resonant Column Test

In order to study how shear modulus and damping ratio depend on shear strain in soil reinforced with SA and cement, a resonant-column test was performed indoors. The test finds the resonant frequency by applying torque in the upper part of the frequency range. The applying power was increased gradually from a low value and applying torque was repeated until the resonant frequency of the specimen was measured. The shear wave velocity, shear modulus, shear strain, and damping ratio can be obtained using the specifications of the equipment and the specimen [2,31]. Figure 3 shows the Stokoe-type resonant-column apparatus (GDSRCA, GDS Instruments; Hook, Hampshire, UK) used in this study. Cylindrical specimens were molded to a height of 100 mm and a diameter of 50 mm. The bottom of the specimen was fixed to the apparatus; the upper surface was freely mounted.

Table 3 describes the specimens used in the resonant column test. The cement mixing ratio (weight ratio of soil and cement) increased from 0 to 2.5, 5.0, and 7.5%. The SA solution mixing ratio (weight ratio of soil and SA solution) was fixed at 7.5%, and the concentrations of the solution(weight ratio of water and sodium alginate) were 0.67% and 3.34%. When the specimen was molded, the under-compaction method was used so that the whole specimen could have a uniform density, and the age period was 15 days (including the period of cementation and formation of the polymer matrix). The confining pressure of the cement-mixed specimen was 100 kPa. The sodium-alginate-mixed specimens were tested sequentially at 100, 200, and 300 kPa; there was a 30 min rest period after the 100 kPa, 200 kPa test were completed.

### 2.3. Viscosity Measurement and Shear-Failure-Characteristics Test

#### 2.3.1. Viscosity Measurement Test

Cement mixes into ground gaps, then forms ettringite over time and hardens. Biopolymers, with a different viscosity, instead form a biopolymer matrix between the ground gaps. Viscosity is the property of resisting the flow of a fluid. The presence of a highly viscous fluid in the ground affects flow behavior in the pores [32,33,34] and increases the adhesion of soil particles [35].

In this study, a LV-DVE viscometer (AMETEK Brookfield; Middleboro, MA, USA) was used to measure the viscosity of SA solutions (Figure 4).

The spindle was placed in the center of the container and used to give the solution a constant rotation. Concentrations of 0.5, 1, 1.5, 2, 2.5, and 3% were tested to confirm their viscosity characteristics. 

Because the SA solution used in this study shows the behavior of a non-Newtonian fluid, a power-law model was applied to determine its flow characteristics [36] (Equation (1)):(1)$$\tau =A+B\times {\dot{\gamma }}^{n},$$
where *τ* is the shear stress (N/m^2^), $\dot{\gamma }$ is the shear rate (1/s), *A* is the shear stress at zero shear rate, *B* is the consistency index, and *n* is the flow-behavior index.

#### 2.3.2. Shear-Failure-Characteristics Test

Shear-failure characteristics were measured to confirm the failure shape of the SA-reinforced soil. The water content of the specimens was fixed at 7.5%, and the specimens were molded using SA solutions at concentrations of 0.67%, 2.00%, and 3.34% (Figure 5a). Each specimen was cylindrical with a height of 100 mm and a diameter of 50 mm (Figure 5b). First, we applied a compressive force to the upper surface of the specimens at a constant speed to confirm failure shape. Second, we took stress measurements while striking the specimen with a hammer a certain number of times to confirm the cohesion of specimens.

## 3. Results and Discussion

### 3.1. Resonant-Column Test

The resonant-column test according to the concentration of mixed materials were as follows: When the proportion of cement was 0%, the maximum shear modulus was 39.5 MPa; when 2.5%, 59 MPa; when 5.0%, 69.4 MPa; and when 7.5%, 93.5 MPa (Table 4). The shear modulus was normalized by its maximum value and plotted against the strain (Figure 6) The normalized shear-modulus curves were fitted with the Ramberg-Osgood model. In the cement-mixture specimens, the range of strains where the normalized shear-modulus curves were linear gradually decreased as the cement content increased. 

According to the viscosity of the SA solution were as follows: the maximum shear modulus of the SA-mixed specimen was greater than that of the unreinforced specimen. When the concentration of SA solution was 0.67%, the maximum shear modulus increased to 78.4 MPa, but when it was 3.34%, the maximum shear modulus was only 46.5 MPa, i.e., it was less than with the lower concentration. This is because the SA was not uniformly distributed throughout the specimen, as will be discussed in more detail in Section 3.3. Im et al. (2017) [2] found a similar trend in the initial test of gellan-gum-treated specimens, and Arab et al. (2019) showed that the value of the resilient modulus of SA-treated specimens decreased for SA concentrations above 2% [28]. This adverse effect appears to be due to a lack of the cations required for SA to form a cemented gel [28]. In the normalized shear modulus curves, the size of the linear region of the SA-mixture specimens decreased compared to that of the unreinforced specimen, but there was no difference between the concentrations. Therefore, the concentration has a mixed effect on the maximum shear modulus, but the reduction ratio of the shear modulus to the shear wave velocity due to resonance is constant. Thus, considering the concentration and viscosity of the mixing materials is essential when applying the reinforcement method.

Figure 7 shows the damping ratio curve of mixture specimens tested with the resonant-column apparatus. At the same shear strain, the damping ratio increased with the amount of cement mixture, and, accordingly, the curve tended to shift to the left. The damping ratio of the sodium alginate mixture specimen was higher than that of unreinforced specimen.

Figure 8 shows the normalized shear-modulus curves and damping-ratio curves for various confining pressures. As the confining pressure increased, the maximum shear modulus of the SA mixture specimens increased (Table 4). In addition, in the normalized shear-modulus curve, as the confining pressure increased, the linear range increased and moved to the right. As the confining pressure increases, the pores of the soil particles become more compact and the more interconnection occurs in the soil particles. As a result, the shear modulus was increased, and because of more pathways, the less energy was dissipated during wave propagation [2,25,37].

There are many ways to model the dynamic properties of the ground. Among them, the Ramberg-Osgood model, modified to fit the strain modulus of the ground, is known to represent the nonlinear range well [22]. For the damping-ratio curve, a hyperbolic model was used [38]. The Ramberg-Osgood model is represented by Equation (2):(2)$$\gamma =\left(\frac{G}{G_{max}}\right)\cdot \gamma +C\cdot {\left(\frac{G}{G_{max}}\cdot \gamma \right)}^{R}$$
where G is the shear modulus, γ is the shear strain and R and C are fitting parameters obtained through the resonant-column test. As the R values increases, the linear range becomes shortened to a perfectly plastic state [39]. As the value of C increases, the linear elastic range of the strain decreases. Table 5 shows the R and C values of the Ramberg-Osgood model using the results of the resonant-column test. When the cement content increased, the R value increased and the C value gradually increased (Figure 9).

In the case of the SA-mixed sample, there was little difference between the R and C values at 100 kPa depending on the concentration, but, as the confining pressure increased, the C value decreased, indicating that the linear elastic section was less sensitive to strain (Figure 10).

### 3.2. Viscosity Measurement Test

In this study, the viscosity was measured at 25 °C. The shear behavior and viscosity of SA solution according to the concentration were as follows: The viscosity ranged from 169.2 to 270 cP for an SA-solution shear rate of 1%, 1330 cP to 1720 cP for 2%, and 4171 to 6030 cP for 3%. Other values are shown in Figure 11. The viscosity characteristics of the SA solution were not constant depending on the shear rate, and the power-law model was applied to represent the non-Newtonian behavior. The consistency index and flow behavior index of the power-law model are presented in Table 6. All of the SA solutions showed shear thinning behavior in which shear stress increased with increasing shear rate.

### 3.3. Shear-Failure-Mode Test

When the specimen mixed with 0.67% SA solution was pressed with a constant load on its upper surface, the shear failure shape appeared (Figure 12a), and the fracture soil was separated into a homogeneous shape when stress was applied (Figure 12b). In the 3.34% concentration specimen, when the constant load was pressed on the top, the side of the cylindrical shape was broken first, and the top of the specimen was not able to withstand the pressure and collapsed (Figure 12c). In addition, when vertical stress was applied, the specimen was unevenly lumped and destroyed (Figure 12d). This suggests that the viscosity of the SA solution increased rapidly as the concentration of the sodium alginate solution increased, and as a result, the SA solution was not homogeneously distributed over the entire sample, but clustered within the soil pores, causing clogging.

The fragments of the fractured specimen were observed under an optical microscope (Figure 13). SA solution causes soil particles to adhere in the form of a bridge and forms a coated film in the soil pore, thereby improving the strength and dynamic properties of the ground in a way similar to cementation [4,17].

## 4. Conclusions

In this study, the shear modulus and damping ratio, which are the dynamic properties of the existing representative cement reinforcement and the soil containing eco-friendly sodium alginate, were obtained through tests. In the case of cement, the experiment was performed while increasing the content (2.5~7.5%). However, while maintaining the sodium alginate content constant at 7.5%, the experiment was performed while changing the concentration of the sodium alginate solution to 0.67% to 3.34%. The results obtained from the test are as follows:(1)As the cement content in the soil pores increases to 7.5%, the shear modulus of the cement reinforced specimens increases. As the concentration of sodium alginate solution increases to 0.67%, the shear modulus of the soil specimen mixed with sodium alginate increases. As the concentration of sodium alginate solution increases (0.5~3.0%), the viscosity increases (66~6030 cp). Sodium alginate is observed to form a matrix between the pores of the soil, which is believed to be the cause of the increase in strength of the specimen.(2)When comparing the sodium alginate solution mixed specimen and the cement mixed specimen, the maximum shear modulus was higher than that of the 5.05 cement content specimen at sodium alginate solution concentration of 0.67%. This confirmed the possibility of sodium alginate as an eco-friendly ground reinforcement material replacing cement.(3)After that, at a concentration of 3.34%, the shear modulus tends to decrease. The cause of the increase and decrease of the shear modulus of the soil sample can be explained by the distribution pattern of sodium alginate in the pores of the soil. Up to 2% of the sodium alginate solution concentration, the matrix is evenly dispersed in the pores of the specimen, and the shear modulus increases as the concentration increases. However, at concentrations above 3.34%, the matrix is partially agglomerated with the soil in the pores. Rather, it shows a shape of local destruction, so the shear modulus decreases.(4)When mixing the reinforcement material into the soil, there is a difference in the change of the dynamic properties of the ground according to the characteristics of the reinforcement material. When mixing cement and polymer, the maximum shear modulus increases, but it becomes a hardening material that decreases the shear modulus more rapidly as the strain increases. When mixing rubber, clay type, as the shear strain increases, the shear modulus decreases slowly, showing a softening material characteristic. In this study, when cement was mixed, the hardening material characteristics were shown, and sodium alginate also showed the characteristics of the hardening material.(5)The damping increases as the soil pores within the cement content increases to 7.5%. When the sodium alginate solution is mixed, the damping ratio increases, but there is no difference between the concentrations of 0.67% and 3.34%. As the confining pressure increases, the pores between the soil particles are compressed, so that the sodium alginate solution adheres to the soil particles better. Therefore, the shear modulus exhibits a hardening characteristic as the strain increases, and the amount of energy reduction decreases through more energy transfer paths.

## Figures and Tables

**Figure 1 materials-14-02743-f001:**
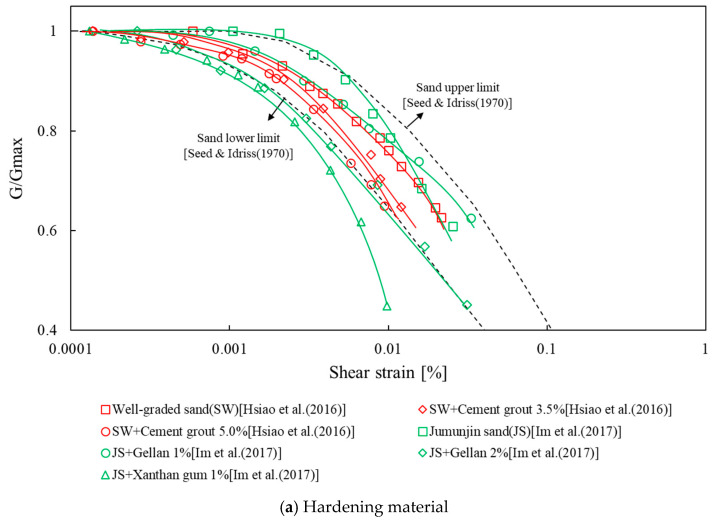
Previous studies of the dynamic characteristics of treated soil: (**a**) Hardening material; (**b**) Softening material; (**c**) Damping ratio. G: shear modulus.

**Figure 2 materials-14-02743-f002:**
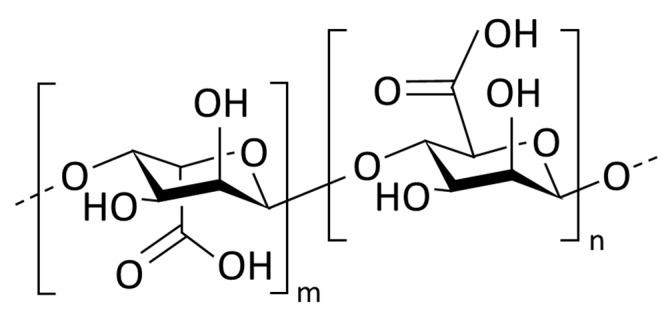
Molecular structure of sodium alginate Reprinted with permission from ref. [30]. Copyright 2017 Springer Nature.

**Figure 3 materials-14-02743-f003:**
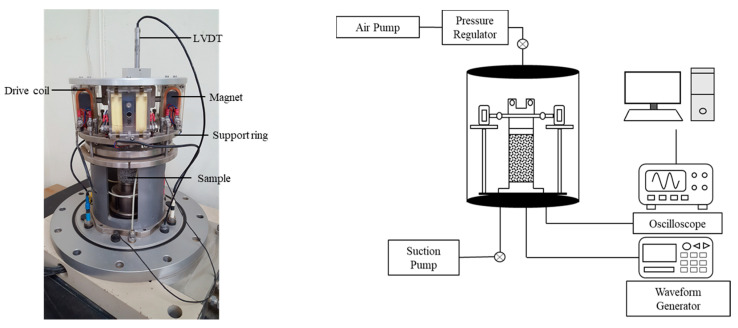
Experimental setup for resonant column test of a soil sample.

**Figure 4 materials-14-02743-f004:**
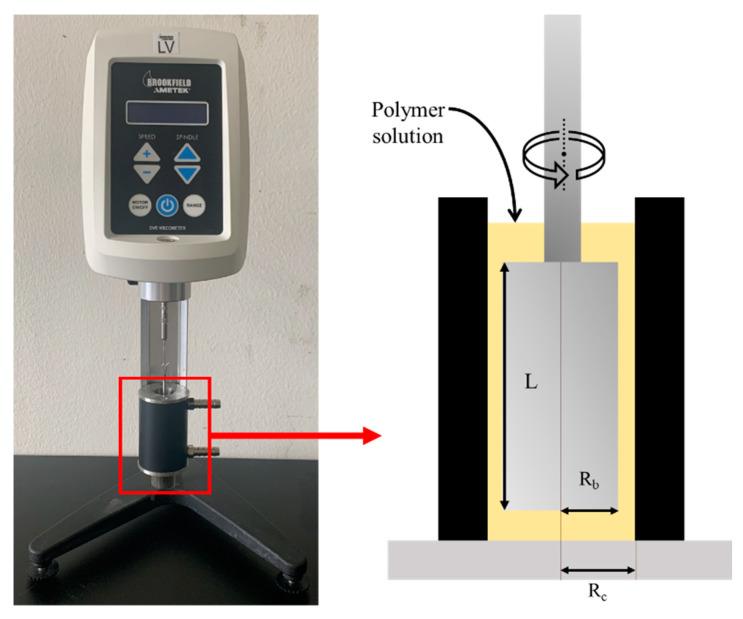
Experimental setup for the measurement of viscosity.

**Figure 5 materials-14-02743-f005:**
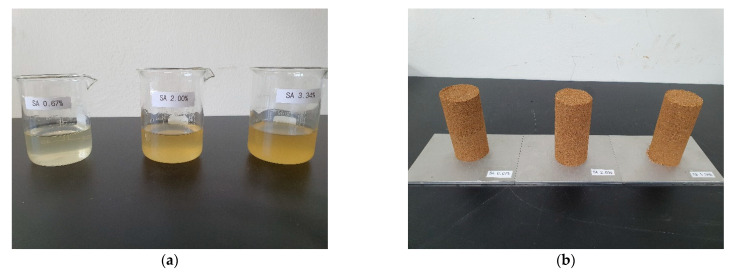
(**a**) Sodium alginate (SA) solutions; (**b**) Specimens after mixture with SA solutions.

**Figure 6 materials-14-02743-f006:**
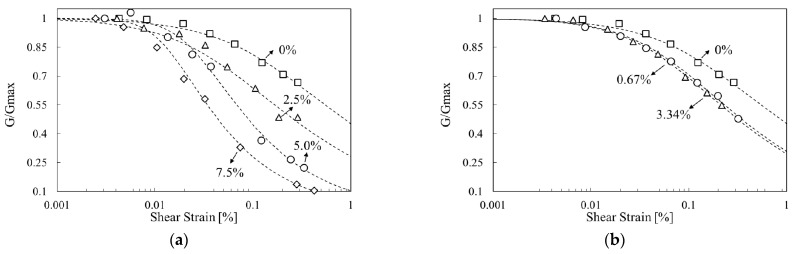
Variation of normalized shear modulus G/G_max_ with shear strain: (**a**) reinforced soils with cement, (**b**) reinforced soils with sodium alginate (SA) solution.

**Figure 7 materials-14-02743-f007:**
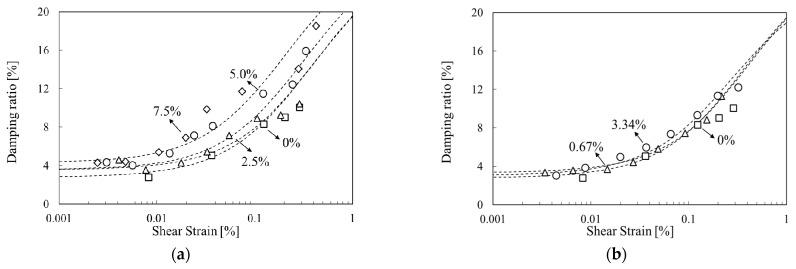
Variation of damping ratio with shear strain: (**a**) reinforced soils with cement, (**b**) reinforced soils with sodium alginate (SA) solution.

**Figure 8 materials-14-02743-f008:**
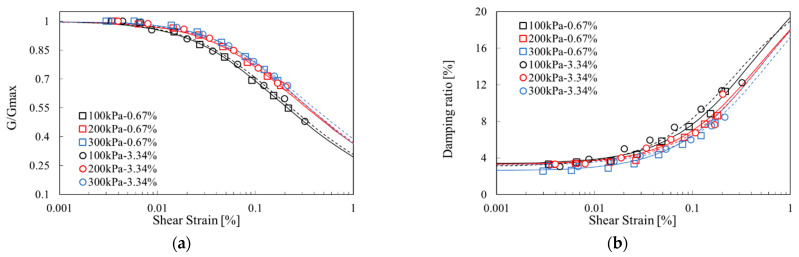
Effects of confining pressure on normalized shear modulus damping ratio: (**a**) relations between normalized shear modulus G/G_max_ and shear strain, (**b**) relations between damping ratio and shear strain.

**Figure 9 materials-14-02743-f009:**
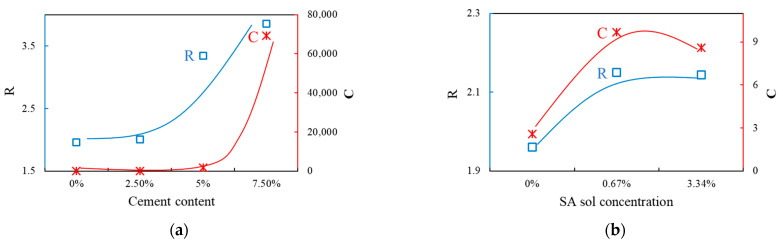
Variation of Ramberg-Osgood model parameters R and C versus (**a**) cement content, (**b**) sodium alginate (SA) solution concentration.

**Figure 10 materials-14-02743-f010:**
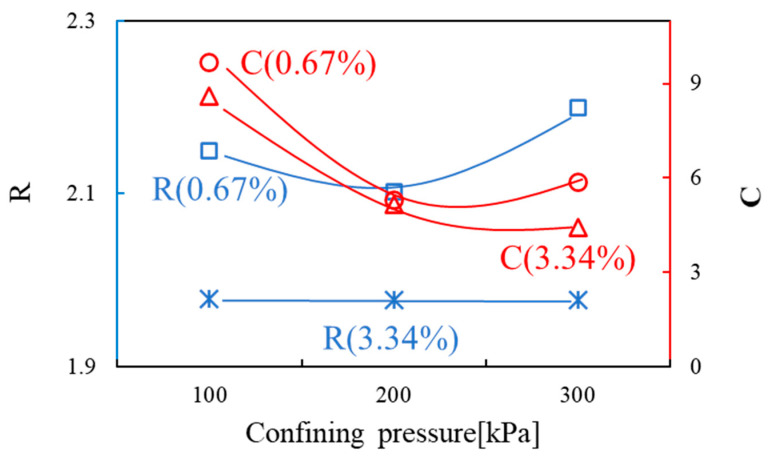
Ramberg-Osgood model parameters R and C versus confining pressure, for sand mixed with various concentrations of sodium alginate solution.

**Figure 11 materials-14-02743-f011:**
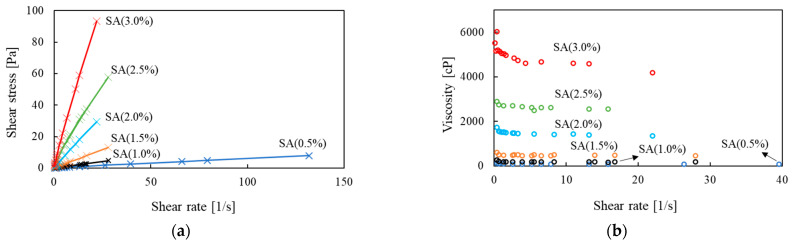
Shear rate of various sodium alginate (SA) solution concentrations versus (**a**) shear stress, (**b**) viscosity.

**Figure 12 materials-14-02743-f012:**
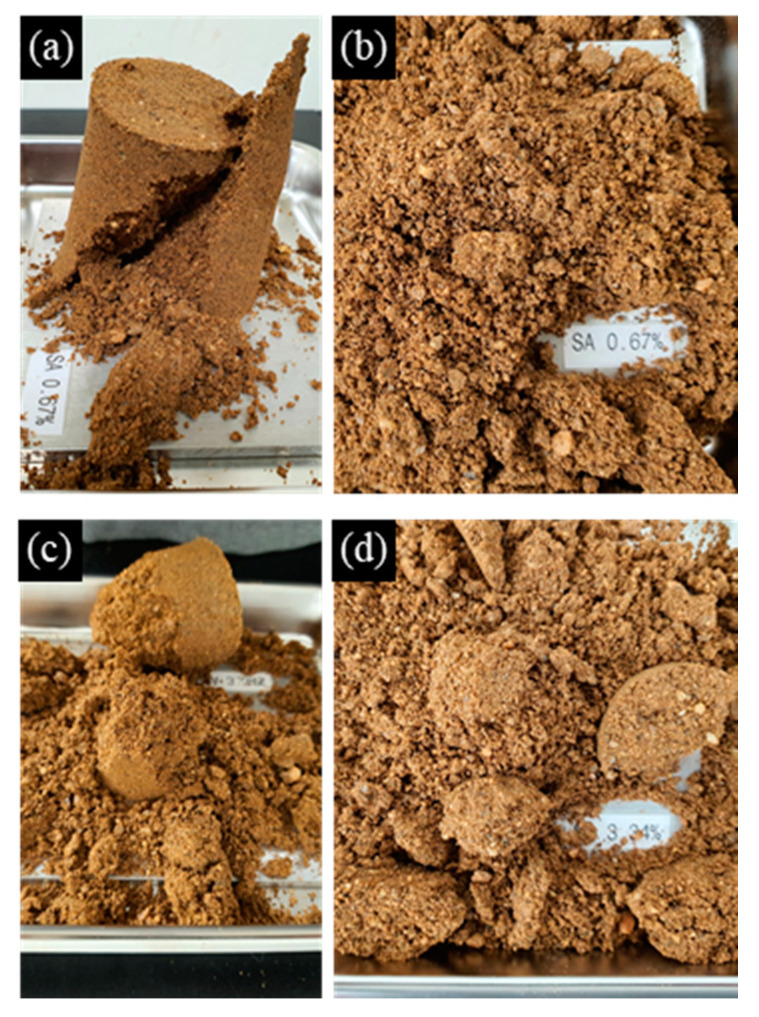
Fracture shape of cylindrical 0.67% (**a**,**b**) and 3.34% (**c**,**d**) sodium alginate (SA) reinforced soil samples after pressure on top surface (**a**,**c**) and subsequent stress (**b**,**d**).

**Figure 13 materials-14-02743-f013:**
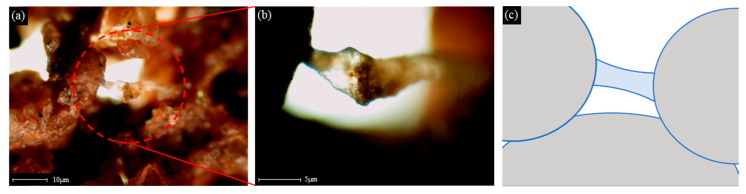
Optical microscope image of sodium alginate reinforced soil particles; (**a**) 4× magnification image of sodium alginate and soil linking; (**b**) 10× magnification image of sodium alginate and soil linking; (**c**) Schematic model of sodium alginate solution treated soil.

**Table 1 materials-14-02743-t001:** Properties of weathered residual soil.

Index		Value for Sample
Sieve Analysis	D_10_	0.18
	D_30_	0.45
	D_60_	1.23
	C_u_	6.67
	C_C_	0.94
	Passing No.4 sieve [%]	100
	Passing No.200 sieve [%]	2.62
Specific gravity		2.61
Unit weight [g/cm^3^]		1.77
USCS		SP

**Table 2 materials-14-02743-t002:** Material properties of sodium alginate sample.

Index	Sample
Formula	(C_6_H_7_O_6_Na)_n_
Source	Macrocystis pyrifera
Moisture	15%
Heavy Metals	0.002%
pH	6.5

**Table 3 materials-14-02743-t003:** Specimens in resonant-column test.

No.	Mixing Materials	Content [%]	Confining Pressure [kPa]
1	Cement	0	100
2		2.5	
3		5.0	
4		7.5	
5	Sodium alginate solution	0.67	100
6			200
7			300
8		3.34	100
9			200
10			300

**Table 4 materials-14-02743-t004:** Shear modulus of reinforced soil in resonant-column test. G_max_: maximum shear modulus.

**Mixing Material**	**Content [%]**	**Density [g/cm^3^]**	**G_max_ [MPa]** **Confining Pressure [kPa]: 100**		
Cement	0	1.77	39.51		
	2.5	1.75	59.10		
	5.0	1.77	67.32		
	7.5	1.81	93.51		
**Mixing Material**	**Content [%]**	**Density [g/cm^3^]**	**G_max_ [MPa]**		
			**Confining Pressure [kPa]**		
			**100**	**200**	**300**
SA solution	0.67	1.69	78.45	89.99	102.01
	3.34	1.79	46.54	57.13	69.56

**Table 5 materials-14-02743-t005:** Ramberg-Osgood model parameters.

Mixing Material	Content [%]	R	C	Confining Pressure [kPa]
Cement	0	1.960	2.580	100
	2.5	2.007	9.040	100
	5.0	3.341	1758.714	100
	7.5	3.857	69,150.070	100
SA solution	0.67	2.150	9.686	100
		2.103	5.298	200
		2.200	5.859	300
	3.34	2.143	8.611	100
		2.121	5.166	200
		2.107	4.425	300

**Table 6 materials-14-02743-t006:** Fitting parameters using power-law model for various SA solution concentrations.

Material	Concentration (%)	A (Y-Intercept)	B (Consistency Index)	n (Flow Behavior Index)	Behavior
SA	0.5	0.0000	0.0859	0.9269	Shear thinning
	1	0.0000	0.1665	0.9807	
	1.5	0.0000	0.4889	0.9836	
	2	0.0000	1.5759	0.9474	
	2.5	0.0000	3.5817	0.8339	
	3	0.0000	5.8144	0.8979	

## Data Availability

Please contact to corresponding author.

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
