# Peer review of "Evaluation of Dynamic Properties of Sodium-Alginate-Reinforced Soil Using A Resonant-Column Test"

_materials, 2021, doi:10.3390/ma14112743_

Round 1
Reviewer 1 Report
This paper carried out an experimental research on the evaluation of dynamic properties of sodium-alginate-reinforced soil using a two resonant-column test. Shear modulus and shear stress were examined by using viscometer and other experimental approaches. Although the topic in this work was interesting, the explanation of the obtained results can be improved. In addition, statements used to explain the obtained results should be reinforced with relevant references. I have detailed my comments below for the sake of the improvement of the manuscript:
- Line 27 typo : "sesmic"
- Line 56 grammatical error, have "been" studied
- Line 73, "Also, ..." (comma)
- Line 77, the phrase is ambiguous, not clear " ... reinforcement increase..." - Distinctive properties of sodium alginate were not given in the introduction with sufficient references.
- Line 86, the sentence is grammatically poor.
- Figure3, oscilloscope (typo)
- Line 127 "minute" (typo)
- Line 136 [29-31]
- The authors should explain more in detail how did they find out the shear modulus from the resonant column tests. References or expressions should be given.
- The root causes of statement given between lines 202-208 should be given, use references.
- The authors can reinforce their statement given in lines 265 - 268 with relevant references if any available.
- The conclusions should be more than a summary of the all obtained results. They should also indicate a direction for the future studies as well as the short useful comments on the obtained results.
Author Response
"Please see the attachment."

Reviewer 2 Report
Dear authors,
congratulations for the work you have done.
The editing of the paper was done in a hurry, please respect the format you have been asked by the publisher.
A few corrections:
line 56 - change the order of references
Figure 1 a and b, remove the brackets from the y axis,
Table 1, use the same precision for D60 and Cc
Figure 3, mark the components on the photo of device
line 136, close the bracket after the reference
How was chosen the power-law model to determine the flow characteristics?
Lines 164-165, line 172-173, 236-238, 250- please reformulate the phrases. The results ...
line 214 - correct to (2)
Figure 12, make a), b) c) and d) visible on the pictures and use as references the letters in the caption of figure, instead of upper row ...
Figure 13 mark the pictures with a) b) and c). Mark the scale on the photos. Please specify in the caption of the figure what represent in the caption of the figure each picture.
Lines 326-330 are necessary?
Author Response
"Please see the attachment."

Reviewer 3 Report
The following comments and suggestions will improve the understanding of the material presented by this paper.
Review comments
- There are some English language corrections that have to be made. For example, in line 57 ‘…and biopolymer have also studied…’ should be corrected to ‘…have been studied..’; line 113 ‘…be giving the torsion…’ should be corrected (for example ‘…by applying torque…’; line 127 ‘…30-munute rest period…’ should be corrected to ’30-minute…’, etc.
- The authors used poorly graded sand for their investigation. Can other types of soil benefit from SA reinforcement?
- Lines 120-123 have to be written clearer so someone can understand what the concentration is referred to, the mixing ratio, etc.
- The authors used two SA solution concentration (0.67% and 3.34%). There is a big difference between the first and the second considering that the 0.67% is more beneficial than the 3.34%. Up to what SA solution concentration the soil can be benefitted?
Author Response
"Please see the attachment."

Round 2
Reviewer 1 Report
The authors have thoroughly corrected/modified reviewers recommendations. However,
The paragraph before Figure 8 was not corrected/modified. The explanations are still missing.
Conclusions are still in the form of a summary.
